# Cellular evolution of the hypothalamic preoptic area of behaviorally divergent deer mice

Jenny Chen[1,2]*, Phoebe R Richardson[2], Christopher Kirby[2], Sean R Eddy[1,3], Hopi E Hoekstra[1,2]*

[1]Department of Molecular & Cellular Biology, Harvard University, Cambridge, United States; [2]Department of Organismic and Evolutionary Biology, Harvard University, Cambridge, United States; [3]Howard Hughes Medical Institute, Harvard University, Cambridge, United States

*For correspondence:
jennifer_chen@fas.harvard.edu
(JC);
hoekstra@oeb.harvard.edu
(HEH)

**Competing interest:** The authors declare that no competing interests exist.

## eLife Assessment

This **important** study identifies species- and sex-specific neuronal cell types and gene expression in the preoptic area (POA) to help understand the evolutionary divergence of social behaviors. The evidence from single-nucleus RNA sequencing and immunostaining is **compelling** and suggests that cellular differences in the POA may contribute to behavioral variations such as mating and parental care that are apparent in two closely related deer mouse species. These rich observations provide an entry point for future hypothesis-driven experiments to demonstrate a causal role for these populations in sex- or species-variable behaviors in vertebrates. These data will be a resource that is of value to behavioral neuroscientists.

**Abstract** Genetic variation is known to contribute to the variation of animal social behavior, but the molecular mechanisms that lead to behavioral differences are still not fully understood. Here, we investigate the cellular evolution of the hypothalamic preoptic area (POA), a brain region that plays a critical role in social behavior, across two sister species of deer mice (*Peromyscus maniculatus* and *P. polionotus*) with divergent social systems. These two species exhibit large differences in mating and parental care behavior across species and sex. Using single-nucleus RNA-sequencing, we build a cellular atlas of the POA for males and females of both *Peromyscus* species. We identify four cell types that are differentially abundant across species, two of which may account for species differences in parental care behavior based on known functions of these cell types. Our data further implicate two sex-biased cell types to be important for the evolution of sex-specific behavior. Finally, we show a remarkable reduction of sex-biased gene expression in *P. polionotus*, a monogamous species that also exhibits reduced sexual dimorphism in parental care behavior. Our POA atlas is a powerful resource to investigate how molecular neuronal traits may be evolving to give rise to innate differences in social behavior across animal species.

## Introduction

Large, heritable differences in complex behavior are observed even across closely-related species and across sexes within the same species. What are the molecular substrates, encoded by genes, that facilitate this variation in innate animal behavior? Two molecular traits known to govern behavior are: (1) abundances of specific neuronal cell types, which may be controlled by developmental genes involved

**eLife digest** Animals display an astounding variety of behaviors that they perform instinctively without needing prior experience. But how does the brain evolve to give rise to these behaviors? Two closely-related species of deer mice found in the United States display very different instinctive behaviors for mating and parental care. The individuals of one species mate with lots of different partners (referred to as promiscuous) and only the females care for the offspring. On the other hand, mice of the other species only have one mate at a time (monogamous) and share the parental care.

Previous studies have shown that a region of the brain called the preoptic area of the hypothalamus regulates mating, parental care and other instinctive behaviors. However, it remains unclear whether there are any differences in the brain cells of the monogamous and promiscuous deer mice that may account for the differences in their behavior. Chen et al. used an approach called single-nucleus RNA-sequencing to compare the cells in the preoptic area of the hypothalamus in these two species.

The experiments identified four types of cells that are present at different levels in the two species. For example, the monogamous deer mice had more cells that express a protein known as galanin, which is known to have a role in regulating parenting behavior in mice, frogs and fish. Furthermore, the cells of the monogamous deer mice, also had fewer "sex-biased" genes, that is, genes that are expressed at different levels in males and females, compared to the promiscuous species.

In both species, two types of cells that are able to sense sex hormones including estrogen and androgen were more abundant in males than females. These cells express many genes that are sex-biased indicating they may play roles in regulating male- or female-specific behaviors.

This work provides a large set of data for further investigations into how genes control behavior and evolve in deer mice and other animals. In the future, this may enable researchers to identify specific genes governing instinctive social behaviors, which may in turn lead to new diagnostic tools and treatments for neurodevelopmental diseases and neuropsychiatric disorders.

in neurogenesis and apoptotic pathways, and (2) abundances of proteins within neuronal cells, which may be controlled by genes involved in transcriptional and translational regulation.

Variations in both types of traits have been documented to change behavior across species and sex. In *Drosophila*, loss of song-patterning neurons underlies a species-specific loss of a type of courtship song (*Ye et al., 2024*) while the expansion of an olfactory neuronal population contributes to *D. sechellia*'s species-specific specialization for feeding on the noni fruit (*Auer et al., 2020*). In mammals, neuronal number differences mediating behavioral traits have largely been described across sexes rather than species. Neuronal population sizes, particularly of cell types within the preoptic area and bed nucleus of the stria terminalis, are thought to be regulated by sex-specific levels of neuroestrogens that control apoptosis, thereby leading to sex-biased cell abundances and sex-specific control of behavior (*Amateau and McCarthy, 2004*; *Tsukahara and Morishita, 2020*).

Across species, gene expression level differences of neuronally-important proteins such as neurotransmitter receptors, smell and taste receptors, and protein kinases are also known to cause behavioral differences (see *Bendesky and Bargmann, 2011* for review). For example, the expression level of the vasopressin 1 a receptor gene in the ventral pallidum contributes to the evolution of partner preference formation in monogamous voles (*Young et al., 1997*; *Young et al., 1999*). Across sexes, differential expression of sex-steroid receptors in the mammalian brain is well-known (*Gegenhuber and Tollkuhn, 2020*), and the resulting consequences for sex-specific gene expression and control of sex-specific behavior is an area of heavy investigation (*Knoedler et al., 2022*; *Gegenhuber et al., 2022*; *Wei et al., 2018*; *Karigo et al., 2021*).

Advances in single-cell and single-nucleus RNA-sequencing have enabled high-throughput characterization of both cell type abundance evolution and cell type-specific expression evolution, and recent single-cell profiling of the brain of behaviorally-divergent species have revealed intriguing insights about both modes of evolution (*Tosches et al., 2018*; *Shafer et al., 2022*; *Johnson et al., 2023*). However, the majority of comparative single-cell RNA-sequencing in mammals has been performed across fairly diverged species with many phenotypic differences, such as human and mouse ~90 million years diverged (*Kumar et al., 2017*) or across primates (*Welch et al., 2019*; *Shekhar et al., 2016*; *Bakken et al., 2021a*; *Khrameeva et al., 2020*; *Bakken et al., 2021b*) such as humans

and marmosets (~40 million years diverged [*Kumar et al., 2017*]), lending limited insight into which of the specific behavioral difference may be affected by which of the many observed molecular changes.

Closely-related species of *Peromyscus* deer mice that exhibit variation in heritable behaviors give us an opportunity to more precisely dissect the molecular neuronal mechanisms that can alter innate mammalian behaviors (*Bedford and Hoekstra, 2015*; *Dewey and Dawson, 2001*). In particular, *P. maniculatus* and *P. polionotus* diverged only ~2 million years ago (*Schenk et al., 2013*), but lie on opposite ends of the monogamy-promiscuity spectrum and exhibit large, heritable differences in their parental care behavior (*Foltz, 1981*; *Bendesky et al., 2017*). Both males and females of the monogamous *P. polionotus* are more parental than the promiscuous *P. maniculatus* (*Foltz, 1981*; *Bendesky et al., 2017*). Further, this difference is much more distinct across males: while promiscuous *P. maniculatus* fathers exhibit little to no parental care in laboratory assays, monogamous *P. polionotus* fathers provide care to nearly the same level as *P. polionotu*s mothers (*Foltz, 1981*; *Bendesky et al., 2017*). These differences provide an opportunity to identify the molecular products that evolve to modulate social behavior, and how they do so in a sexually dimorphic manner.

Evolution of the hypothalamus has been implicated in social behavior differences across many species (*Xie and Dorsky, 2017*; *Alié et al., 2018*; *Kanda, 2019*), including across these two species of *Peromyscus* (*Bendesky et al., 2017*). Here, we used single-nucleus RNA-sequencing (snRNA-seq) to profile the POA, a region of the hypothalamus known for its many cell types involved in mating and parenting behavior (*Mei et al., 2023*), amongst other innate behaviors. We profiled males and females of both *P. maniculatus* and *P. polionotus* species and, by comparing to *Mus* data, identified POA cell types conserved across rodents. We identified cell type abundance and gene expression differences across species and sex, and hypothesized how these differences may contribute to species- and sex-specific behaviors. Finally, we characterized how sex-bias is evolving across these two *Peromyscus* species and proposed potential cell type abundance and gene expression changes that may be co-evolving with monogamy.

## Results
### A transcriptional cell atlas of *Peromyscus* POA
We performed droplet-based snRNA-seq on the POA of 6 males and 6 females from each of the *P. maniculatus* and *P. polionotus* species (Methods) (*Figure 1A*). An atlas of the POA in *Mus musculus* was previously developed that combined single-cell RNA-seq data with imaging-based spatial information and data on specific neuronal populations activated by social behaviors (*Moffitt et al., 2018*). To create a comparable dataset, we used a dissection strategy identical to that used for creating the *Mus* POA atlas to dissect a region containing the POA and surrounding nuclei (~2.5 mm × 2.5 mm × 0.75 mm equivalent to *Mus* Bregma +0.5 to –0.6) (Methods, *Figure 1B*). To reduce batch effect, thereby increasing statistical power for detecting species and sex differences, we took advantage of the genetic diversity of our outbred study system and pooled nuclei from a male and female from each of the *P. maniculatus* and *P. polionotus* species (i.e. 4 samples) into each snRNA-seq run (*Figure 1C*). We collected a total of 105,647 nuclei across 6 replicates (i.e. 24 samples) (Methods, *Supplementary file 1*). To computationally demultiplex the pooled data, we used bulk RNA-seq or whole genome sequencing data to call single nucleotide polymorphisms (SNPs) for each animal. We then used SNP data to assign sample information to each nucleus in the snRNA-seq dataset (Methods, *Figure 1— figure supplements 1–3*). In total, we were able to confidently assign a single sample to 95,057 nuclei. The remaining 10,590 nuclei appeared to be multiplets containing two or more samples and were removed from subsequent analyses.

We combined data across all samples and clustered the data using Seurat v4.1.3 (*Hao et al., 2021*) (Methods). We first performed a low-granularity clustering using the first 30 principal components (PCs) of our data and a low resolution parameter of 0.2 to obtain a small number of clusters. This procedure identified 21 major cell classes which were assigned as neurons, microglia, astrocytes, immature and mature oligodendrocytes, endothelial, and ependymal cells using known marker genes from *Mus* hypothalamus (*Steuernagel et al., 2022*; *Figure 1—figure supplement 4*). We then focused only on neuronal nuclei (52,121 nuclei), and reclustered the data at a higher granularity using the first 100 PCs of our data and a resolution parameter of 1.8 (Methods). This resulted in 53 neuronal cell clusters (*Figure 1D*). We did not find major biases in sex or species across our clusters, as shown by the even

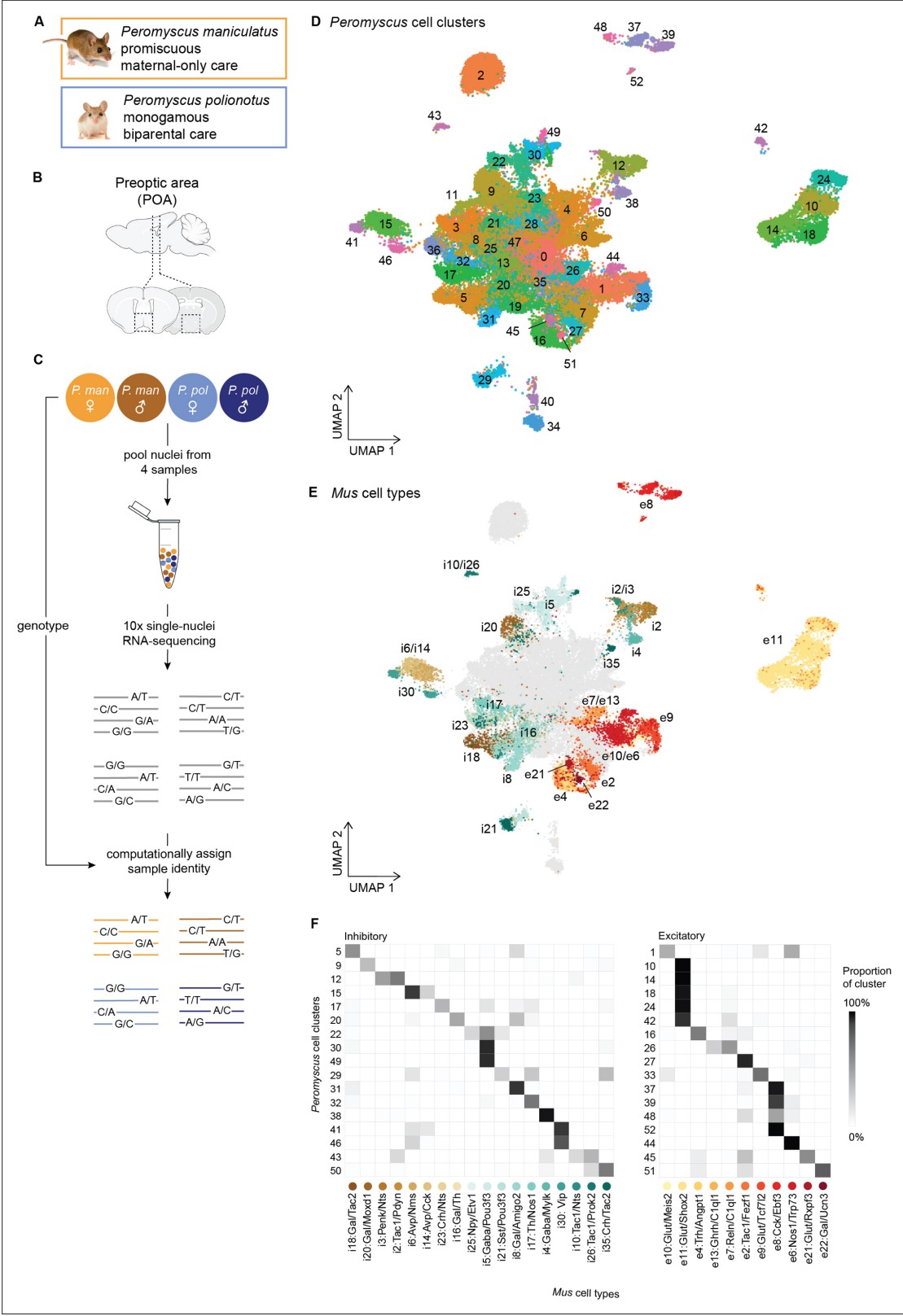

**Figure 1.** Overview of *Peromyscus* hypothalamus single-cell atlas. (**A**) Images of *Peromyscus maniculatus* and *Peromyscus polionotus* and summary of their social behavioral differences. (**B**) Schematic of the preoptic area of the hypothalamus. Dotted boxes indicate the area dissected. (**C**) Schematic of pooling and sample assignment strategy used for single-nucleus RNA-sequencing (snRNA-seq). (**D**) UMAP visualization of all 52,121 neuronal nuclei collected across 6 males and 6 females from each of the *P. maniculatus* and *P. polionotus* species, colored by cell cluster. (**E**) UMAP visualization of all

*Figure 1 continued on next page*

*Figure 1 continued*

neuronal nuclei, colored by the assigned *Mus* cell type label. Only *Peromyscus* clusters for which a homologous *Mus* cell type label could be found are colored; remaining nuclei are shown in gray. (**F**) Heatmap of the proportion of each *Peromyscus* cell cluster (rows) that are assigned to each *Mus* cell type label (columns). Only *Peromyscus* clusters for which a homologous *Mus* cell type label could be found are shown.

Image credit: Joel Sartore / Photo Ark.

The online version of this article includes the following source data and figure supplement(s) for figure 1:

**Figure supplement 1.** Schematic illustrating pipeline for singlet and multiplet assignment of pooled single-nucleus RNA-sequencing (snRNA-seq) data.

**Figure supplement 2.** Sample assignment for pooled RNA-seq data.

**Figure supplement 2—source data 1.** Related to *Figure 1—figure supplement 2*.

**Figure supplement 3.** Statistics of demultiplexed single-nucleus RNA-sequencing (snRNA-seq) data.

**Figure supplement 3—source data 1.** Related to *Figure 1—figure supplement 3*.

**Figure supplement 4.** Clustering of single-nucleus RNA-sequencing (snRNA-seq) data into major cell classes.

**Figure supplement 5.** UMAP visualizations of all neuronal nuclei, separated by sex and species.

**Figure supplement 6.** Expression level and spatial distribution of six known cell type-specific genes.

**Figure supplement 7.** Classification of inhibitory and excitatory cell clusters.

**Figure supplement 8.** Homology mapping of *Mus* cell type labels to *Peromyscus* cell clusters.

**Figure supplement 8—source data 1.** Related to *Figure 1—figure supplement 8*.

**Figure supplement 9.** Sequencing metrics across cell clusters.

**Figure supplement 9—source data 1.** Related to *Figure 1—figure supplement 9*.

distribution of each sex and species across the UMAP (*Figure 1—figure supplement 5*). Our cell clusters reflect known cell types of the POA as well as surrounding nuclei of the anterior hypothalamus, including the paraventricular (PVN), supraoptic (SON), suprachiasmatic nuclei (SCN), and more. For example, arginine vasopressin (*Avp*) and oxytocin (*Oxt*) are exclusively and highly expressed in clusters 34 and 40 (*Figure 1—figure supplement 6*), respectively, and reflect the well-known populations of AVP and OXT neurons that are found in the PVN and SON. Additionally, vasoactive intestinal protein (*Vip*) is highly expressed in clusters 41 and 46 (*Figure 1—figure supplement 6*), which together represent the VIP neurons of the SCN that play a major role in regulating circadian rhythms. Further inspection of these two clusters finds specific expression of neuromedin S (*Nms*) only in cluster 41 and gastric releasing peptide (*Grp*) only in cluster 46 (*Figure 1—figure supplement 6*), suggesting that these two clusters actually represent two subpopulations of VIP neurons, Vip/Nms and Vip/Grp, that have recently been reported as genetically and functionally distinct cell types (*Todd et al., 2020*; *Wen et al., 2020*).

While some neuropeptidergic cell types were obviously identifiable in our data, they were the minority of our cell clusters. We sought to provide more biological context for the rest of our data by identifying homology between our cell clusters and cell types defined from the *Mus* POA atlas (*Moffitt et al., 2018*). We used Seurat's integration workflow (*Stuart et al., 2019*) to identify conserved gene correlation structures across the *Mus* and *Peromyscus* datasets and transfer *Mus* cell type labels onto each *Peromyscus* nucleus (Methods, *Figure 1E*). Because the *Mus* POA atlas consists of separate inhibitory and excitatory datasets, we first defined each of our nuclei as inhibitory or excitatory based on whether it expressed higher levels of inhibitory markers *Gad1* and *Gad2* or excitatory marker *Slc17a6* (Methods, *Figure 1—figure supplement 7*). We then transferred inhibitory and excitatory *Mus* cell type labels only onto inhibitory and excitatory nuclei, respectively.

For each *Peromyscus* cluster, we calculated the proportion of nuclei within the cluster assigned to each *Mus* cell type and labeled clusters if at least 15% of the cluster was predicted to belong to the same *Mus* cell type label (Methods). This procedure identified mappings for 34 of our 53 neuronal cell clusters (*Figure 1E, F*, *Figure 1—figure supplement 8*). (*Mus* cell type labels were denoted by *Moffitt et al., 2018* as inhibitory (i) or excitatory (e), and named after marker genes identified from *Mus* POA atlas.) The cell clusters for which we were able to assign *Mus* homology include four cell types activated by relevant social behaviors in *Mus*: cluster 5 (i18: Gal/Tac2), which is activated by mating behaviors; cluster 17 (i23: Crh/Nts) which is activated by parenting behaviors; and clusters 20 (i16: Gal/Th) and 9 (i20:Gal/Moxd1), which are activated by both mating and parenting behaviors (*Moffitt et al., 2018*).

We were unable to assign *Mus* homology for 19 *Peromyscus* cell clusters. These cell clusters had similar numbers of reads and genes per nuclei compared to clusters for which we did find *Mus* homology (*Figure 1—figure supplement 9*), and therefore we do not believe the lack of homology assignment is due to technical artifacts. Instead, clusters unassignable to *Mus* cell types likely fall into two categories: First, dissection and/or anatomical differences between *Peromyscus* and *Mus* may influence the presence and absence of certain cell types. For example, cluster 2 forms a clear 'island' on our UMAP and is transcriptionally specified by high expression of *Pvalb*, *Gad1*, and *Gad2* (*Figure 1D*, *Figure 1—figure supplement 6*, *Figure 1—figure supplement 7*). This cluster very likely consists of inhibitory interneurons that have been reported to reside primarily in the thalamus in *Mus* (*Clemente-Perez et al., 2017*) but may have been inadvertently captured in our dataset due to differences in dissection or differences in spatial distribution of these interneurons between species. Second, some cell types may not be transcriptionally well-defined and homology would not be assignable based on gene expression data alone. For example, cell clusters that lie in the middle of our UMAP (e.g. clusters 0, 13, 25, 8, etc.) likely do not have strong transcriptional signatures captured by our snRNA-seq and were also less likely to be mapped to a homologous *Mus* cell type. Still, we were able to assign homology for the majority of our nuclei enabling us to better interpret cellular and molecular changes in the context of their biological functions.

## Differential abundance of cell clusters across species and sex

With our *Peromyscus* POA cell atlas, we first asked if there were differences in cell abundance across species or sex (*Figure 2A*). To do so, we treated cell abundance as count data and used edgeR (*Robinson et al., 2010*) to fit a generalized linear model that included replicate, sex, and species as covariates (*Figure 2A*, *Supplementary file 2*, Methods). Importantly, edgeR controls for differential numbers of neurons sequenced across samples (Methods). We used edgeR to test for significant coefficients (i.e. differential abundance across species or sex) and accounted for multiple hypothesis testing with a false discovery rate (FDR) correction (*Benjamini and Hochberg, 1995*). Because we would later validate our results experimentally, we used a lenient FDR cutoff of 0.3 to initially identify candidate clusters showing differential abundance. We performed this analysis on all cell clusters, regardless of whether they were mapped to a homologous *Mus* cell type or not.

Across our 53 neuronal clusters, we found four clusters that were called differentially abundant across species and two that were called differentially abundant across sexes (*Figure 2B*). Arginine vasopressin (AVP) and oxytocin (OXT) neurons (clusters 34 and 40, respectively) were both ~1.6 x more abundant (AVP FDR = 0.19; OXT FDR = 0.09) in the promiscuous *P. maniculatus* than the *P. polionotus* (*Figure 2C*). The AVP neuron number difference corroborates a previous finding that *P. maniculatus* had significantly more AVP neurons than *P. polionotus* (~1.7 x) in the medial paraventricular nucleus (*Cushing, 2016*). Furthermore, we previously reported based on bulk RNA-seq data that the expression level of *Avp* was nearly 3 x higher in *P. maniculatus* hypothalamus compared to *P. polionotus*, and that this expression difference mediates the different amounts of parental nesting behavior performed by each species (*Bendesky et al., 2017*). Our data suggests this difference in *Avp* expression and parental nesting behavior may be driven, at least in part, by a neuronal number difference across species.

Two cell clusters were found to be more abundant in the monogamous *P. polionotus* compared to the promiscuous *P. maniculatus*: cluster 43 (i26:Tac1/Prok2) (3.4 x higher; FDR = 0.08) and cluster 20 (i16:Gal/Th) (1.6 x higher; FDR = 0.02) (*Figure 2C*). We were intrigued by the latter cluster because of its known roles in parental care. In *Mus*, i16:Gal/Th is activated by pup exposure and successful mating in both sexes (*Moffitt et al., 2018*). This cell type likely represents a subset of the galanin-expressing neurons in the medial preoptic area (MPOA) that were previously reported to elicit pup grooming in *Mus* fathers when optogenetically activated (*Wu et al., 2014*). In *Peromyscus*, we also observed robust and specific expression of galanin in this cell cluster (*Figure 2—figure supplement 1*). Previously, we have shown that *P. polionotus* of both sexes perform more parental care than their *P. maniculatus* counterparts, and that this behavioral difference is heritable (*Bendesky et al., 2017*). Thus, we found a positive association between innate parental care behavior and number of a galanin-expressing neuronal cell cluster across *Peromyscus* species.

Finally, we identified two cell clusters to be more abundant in males than females in both *Peromyscus* species: cluster 9 (i20:Gal/Moxd1) (1.7 x higher; FDR = 0.007) and cluster 5 (i18:Gal/Tac2)

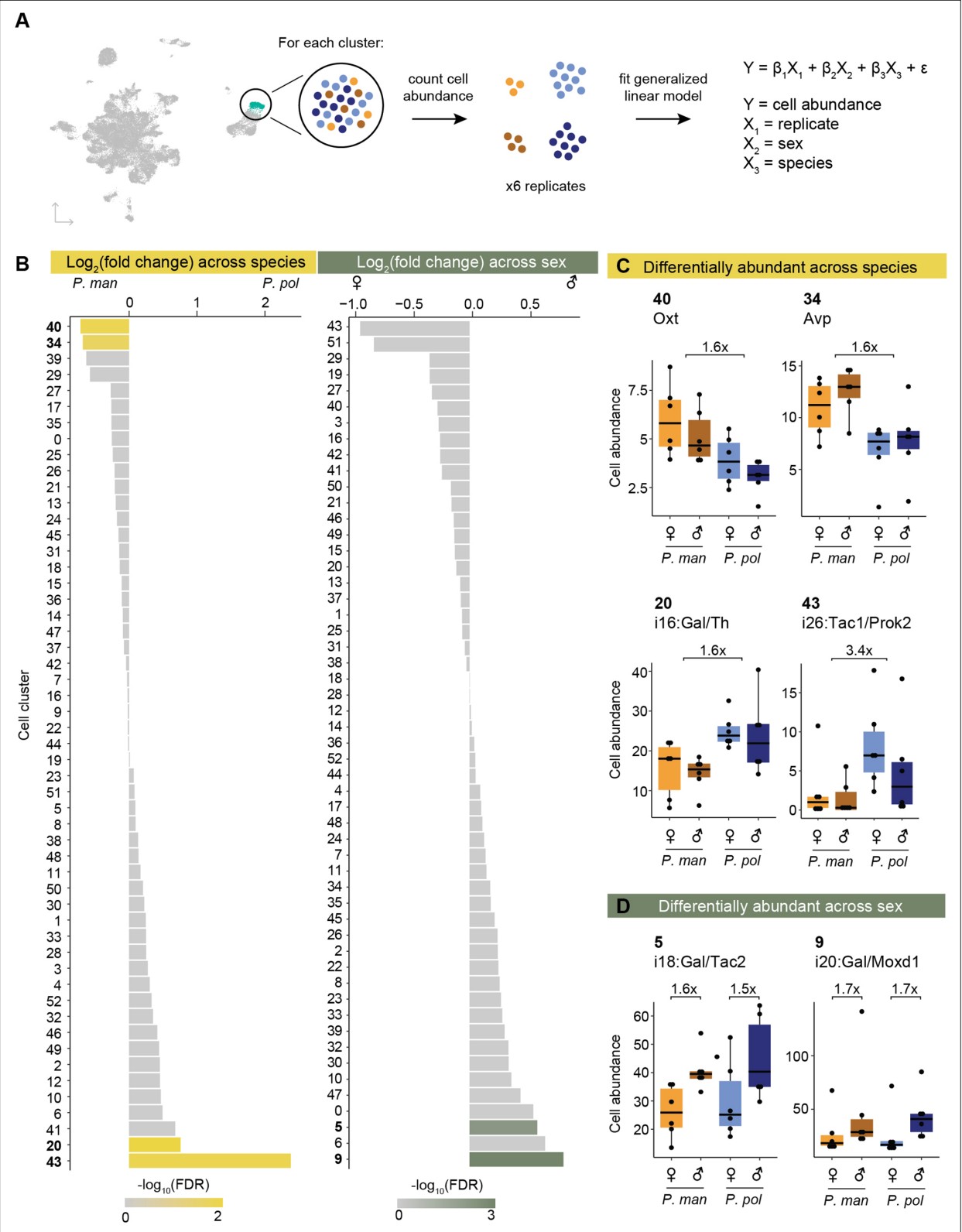

**Figure 2.** Differential abundance of cell types across species and sex. (**A**) Schematic of procedure used to test for differential abundance of cell clusters. (**B**) Barplots of log₂(fold change) across species (*P. pol / P. man*, left) and across sex (male/female, right) for each cell cluster. Bars are colored by -log₁₀(FDR). (**C**) Boxplots of cell abundances (y-axis) of four cell types differentially abundant across species and (**D**) of two cell types differentially abundant across sex. Cell abundances across samples are normalized using TMM normalization (Methods).

*Figure 2 continued on next page*

*Figure 2 continued*

The online version of this article includes the following source data and figure supplement(s) for figure 2:

**Source data 1.** Related to *Figure 2B*.

**Figure supplement 1.** *Galanin* (*Gal*) expression across cell clusters.

(1.5 x higher; FDR = 0.27) (*Figure 2D*). Both cell types are activated by mating behaviors in male *Mus* (*Moffitt et al., 2018*). i20:Gal/Moxd1 neurons are found in the sexually dimorphic nucleus of the preoptic area (SDN-POA) as well as the bed nucleus of the stria terminalis (BNST), both of which have been described to be sexually dimorphic and larger in males of many mammals including rats (*Gorski et al., 1980*), *Mus* (*Brown et al., 1999*), and humans (*Allen and Gorski, 1990*; *Hofman and Swaab, 1989*). Here, we again found a male-bias of i20:Gal/Moxd1 in both a promiscuous and monogamous species.

Several of our differentially abundant cell types have known, highly-conserved gene markers: AVP neurons (marked by AVP), OXT neurons (marked by OXT), and i20: Gal/Moxd1 (marked by calbindin1 [CALB1]) (*Tsukahara and Morishita, 2020*; *Tsuneoka et al., 2017*). We took advantage of this to experimentally validate our single-nucleus data. We additionally were interested in investigating interspecies differences in the spatial distribution of differentially abundant cell types. Therefore, we performed immunoreactive staining across *the entire hypothalamus* (Methods). We then counted cells within hypothalamic subregions and tested for differential abundance in the direction identified from our single-nucleus data.

AVP and OXT neurons are formed from a common developmental lineage (*Nakai et al., 1995*; *Schonemann et al., 1995*) and both primarily reside in the PVN and the SON. Additionally, small numbers of AVP and OXT neurons were also found by immunostaining in the anterior hypothalamus area (AHA) and BNST (*Figure 3—figure supplement 1*). Our counts of AVP-immunoreactive (ir) neurons found 1.5 x more cells in the PVN (*P*=7.0e-4, one-sided Mann Whitney test) and 2.1 x more cells in SON (p=6.0e-6) (*Figure 3A and B*) in *P. maniculatus* compared to *P. polionotus*. Our counts of OXT-ir neurons found 3.1 x more cells in the PVN (p=2.8e-6) and 4.7 x more cells in SON (p=6.2e-5) (*Figure 3C and D*). Therefore, both regional populations of AVP and OXT neurons appear to be contributing to their differential abundance across species.

In contrast, when we examined CALB1-ir cells (marking i20:Gal/Moxd1), we found regional differences in differential abundance. As expected, CALB1-ir cells were found in the SDN-POA and BNST. Because the cell density of CALB1-ir cells is so high in the BNST, we calculated the area of immunoreactivity in the BNST rather than counting cells individually. We note that this metric may include fibers and terminals whose signal may originate from projections from regions outside the BNST. The area of immunoreactivity in the BNST was ~2 x larger in males in both *P. maniculatus* (p=0.004, one-sided Mann Whitney test) and *P. polionotus* species (p=0.03) (*Figure 3E and F*). However, in the SDN-POA, we found that the number of CALB1-ir cells was 1.8 x more abundant in the promiscuous *P. maniculatus* males (p=0.03, one-sided Mann Whitney test), but *not* significantly different across sexes in the monogamous *P. polionotus* (p=0.2) (*Figure 3E and F*). i20:Gal/Moxd1 cells of the SDN-POA and BNST cannot be transcriptionally differentiated (*Moffitt et al., 2018*), and the differential abundance observed in our scRNA-seq data was likely driven primarily by large numbers of i20:Gal/Moxd1 cells of the BNST masking the small number of i20:Gal/Moxd1 cells of the SDN-POA. However, using immunostaining to better capture spatial information, we actually find that the two subregions of i20:Gal/Moxd1 are evolving differently across species: while sex-bias in the BNST has remained conserved, sex-bias in the SDN-POA has become more reduced in the monogamous *P. polionotus*.

In sum, we computationally identified six neuronal cell types of the hypothalamus that have changed in abundance across species or sex, and experimentally confirmed three cell types with abundance changes. Our findings align with previous work: we identified a species difference in AVP neurons, which had been reported to be differentially abundant across our focal species (*Cushing, 2016*), and sex-bias of i20:Gal/Moxd1, which is known to be male-biased across mammalian species. Our findings also identified four neuronal cell types with differential abundance across species, two of which are candidates for underlying species differences in behavior: differential abundance of AVP neurons may be partially responsible for a differential expression level of AVP across species, which we previously implicated in parental nesting behavior (*Bendesky et al., 2017*), and an increased abundance of a

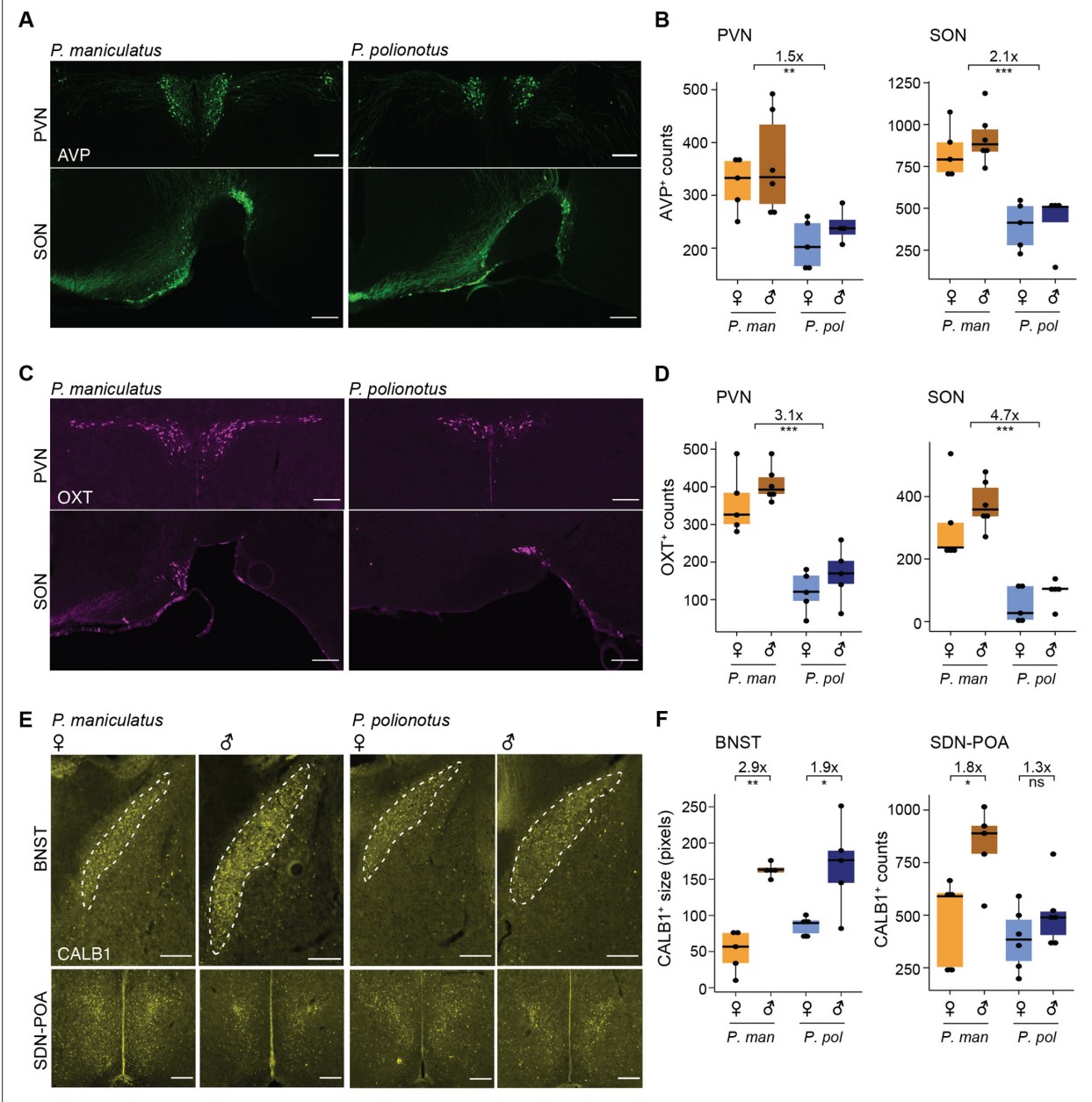

**Figure 3.** Immunostaining and cell counts of regional populations of differentially abundant cell types. (**A**) Representative images of immunoreactive (ir) staining of AVP+ neurons in the paraventricular (PVN) (top) and supraoptic (SON) (bottom) of *P. maniculatus* and *P. polionotus*. Scale bar represents 200 μm. (**B**) Boxplots of AVP+ neuron number counts in the PVN (left) and SON (right) of females and males of *P. maniculatus* (*P. man*) and *P. polionotus* (*P.pol*): *** p<0.001 (one-sided Mann Whitney test). (**C, D**) Same as A and B, but of OXT+ neurons. (**E**) Representative images of ir staining of CALB1+ neurons in the BNST (top) and SDN-POA (bottom) of a female and male *P. maniculatus* and *P. polionotus*. Scale bar represents 200 μm. (**F**) Boxplots of the area of CALB1+ in the bed nucleus of the stria terminalis (BNST) (left) or CALB1+ neuron number counts in the sexually dimorphic nucleus of the preoptic area (SDN-POA) (right) of females and males of *P. maniculatus* and *P. polionotus*: **p<0.01; *p<0.05; ns: not significant (one-sided Mann Whitney test).

The online version of this article includes the following source data and figure supplement(s) for figure 3:

**Source data 1.** Related to *Figure 3B*.

**Source data 2.** Related to *Figure 3D*.

**Source data 3.** Related to *Figure 3F*.

**Figure supplement 1.** Arginine vasopressin (AVP) and oxytocin (OXT) neuron counts across the entire hypothalamus.

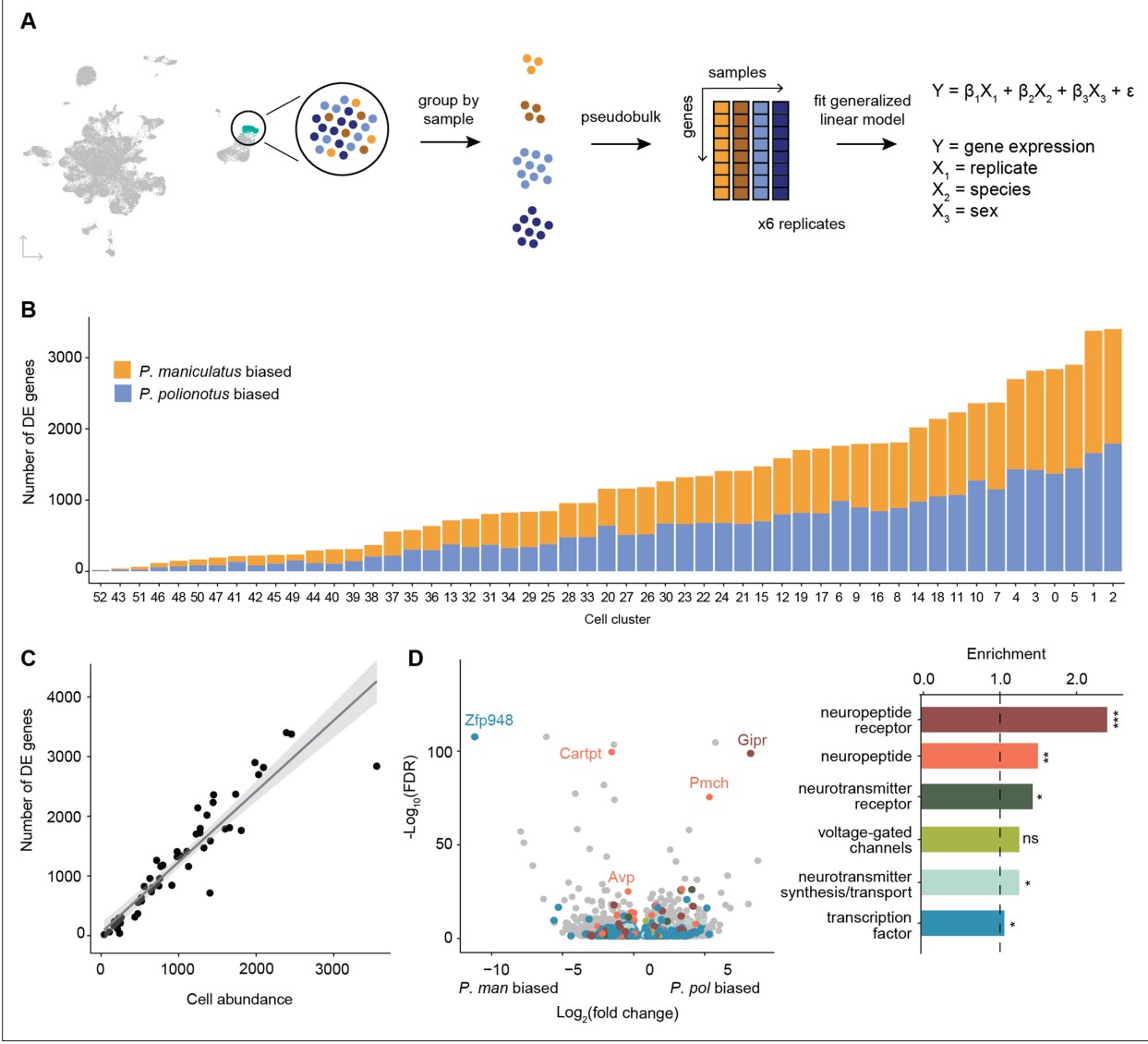

**Figure 4.** Differential gene expression across species. (**A**) Schematic of procedure used to pseudobulk gene expression counts and test for differential gene expression across species. (**B**) Barplot of the number of differentially expressed (DE) genes for each cell cluster (bars) colored by *P. maniculatus* or *P. polionotus* bias. (**C**) Scatter plot of cell cluster abundance (x-axis) and number of differentially expressed genes (y-axis) identified in each cell cluster (dots). Line of best fit (gray line) and 95% confidence interval (gray shading) are shown. (**D**) Left: Volcano plot of $\log_2$(fold change) (x-axis) and $\log_{10}$(FDR) (y-axis) of DE genes across species. Negative values of $\log_2$(fold change) indicate *P. maniculatus* bias and positive values indicate *P. polionotus* bias. For genes DE in more than one cell type, the cell type with the highest expression level is shown. Genes are colored by gene categories listed on the right. Select outlier genes are labeled. Right: Barplot of enrichment scores (Methods) for each gene category. Dotted line at 1.0 indicates no enrichment or depletion. \*\*\*FDR <0.001; \*\*FDR <0.01; \*FDR <0.05; ns: not significant.

galanin-expressing cell type known to govern parental care behavior, i16:Gal/Th, was observed in the monogamous species that performs more parental care. Finally, our immunostaining found that regional populations of transcriptionally identical cell types may evolve differently across species, and suggests that regional-specificity of sex-bias in i20:Gal/Moxd1 cells may be important to the evolution of sex-specific behaviors.

## Differential neuronal gene expression across species and sex

We next asked what gene expression differences there were across species and sex. To do this, we pseudobulked our single-nucleus expression data and summed up gene counts across cells within the same cell cluster and sample (*Figure 4A*). We again used edgeR to fit a generalized linear model that

included replicate, sex, and species as covariates (Methods). Because edgeR normalizes for library size across samples, cell type abundance is controlled for in our differential expression analysis. Again, we performed this analysis on all cell clusters, regardless of whether they were mapped to a homologous *Mus* cell type or not.

Across species, we found a range of 18–3401 differentially expressed (DE) genes in each cell cluster (FDR <0.05), with no bias towards either species (*Figure 4B*, *Supplementary file 3*). The number of DE genes in each cell cluster was highly correlated with the abundance of that cell cluster (*Figure 4C*). This is expected given that larger abundances is analogous to deeper sequencing of a cell population and results in more genes detected and more accurate estimates of gene counts. In total, we found 8301 unique genes to be DE in one or more cell clusters. When compared to a background of expression-matched non-DE genes (Methods), DE genes were highly enriched in neuropeptide receptors (2.4 x enrichment, FDR = 7e-04), and modestly enriched in neuropeptides (1.5 x, FDR = 0.005), neurotransmitter receptors (1.3 x, FDR = 0.02), and voltage-gated channels (1.3 x, FDR = 0.03) (*Figure 4D*). We did not find a significant enrichment of genes involved in neurotransmitter synthesis or transport, and only a small, though marginally significant, enrichment of transcription factors (1.1 x, FDR = 0.04) (*Figure 4D*).

Across sexes, we performed differential expression analysis separately for each species to identify changes in sex-biased expression across species (*Supplementary file 4*). Remarkably, we observed far fewer sex-biased genes in the monogamous species compared to the promiscuous species. We found a range of 0–40 DE genes (FDR <0.05) in *P. maniculatus* cell clusters and 2–27 DE genes (FDR <0.05) across *P. polionotus* cell clusters (*Figure 5A*). In total, we found 204 female- and 194 male-biased genes in *P. maniculatus*; in contrast, only 170 (83%) female- and 70 (36%) male-biased genes were detected in *P. polionotus* (*Figure 5B*). Our data suggest a substantial reduction of sex-biased genes in the monogamous species, driven primarily by a reduction of genes with male-biased expression.

Despite the short evolutionary timescales between our species, we found very few sex-biased genes shared across species (*Figure 5B*). Only eight genes were found to be female-biased in the same cell types in both species, including three known female-specific genes associated with X-inactivation (*Xist*, *Tsix*, and *Jpx*). Eleven genes were found to be conserved in male-bias including *Nrip1* (cluster 5, homologous to i18:Gal/Tac2), a nuclear protein that mediates estrogen signaling (*Nautiyal et al., 2013*), and *Ecel1* (cluster 9, homologous to i20:Gal/Moxd1), a neuronal protease (*Kaneko et al., 2017*) also known to be male-biased in the *Mus musculus* hypothalamus (*Xu et al., 2012*).

In both *P. maniculatus* and *P. polionotus*, two cell types had the most number of sex-biased genes: cluster 5 (i18:Gal/Tac2) and cluster 9 (i20:Gal/Moxd1) (*Figure 5A*). When comparing the number of sex-biased genes detected to cell cluster abundance, clusters 5 and 9 were obvious outliers in the number of sex-biased genes detected (*Figure 5C*). The enrichment of sex-biased genes was apparent in both species, even though the two cell types displayed a large reduction in the number of male-biased genes in the monogamous *P. polionotus*.

Notably, these were the same two cell types we found to be *differentially abundant* across sexes (*Figure 2D*). However, we found the enrichment of sex-biased genes to be true even when we re-performed our analysis on data that was downsampled so that cell abundance was equal across sexes (*Figure 5—figure supplement 1*). Additionally, both clusters exhibited high expression of all four major sex-steroid hormone receptors – estrogen receptor α (*Esr1*), androgen receptor (*Ar*), prolactin receptor (*Prlr*), and progesterone receptor (*Pgr*) (*Figure 5—figure supplement 1*). While none of these sex-steroid hormone receptors were found to be significantly sex-biased in expression after controlling for multiple hypothesis testing of the entire transcriptome (*Supplementary file 4*), individual inspection of these genes reveals subtle patterns of sex-biased expression: *Esr1* is female-biased in both species in cluster 5 while *Ar* is trending towards male-bias in both species cluster 9 (*Figure 5—figure supplement 2*). These receptors may be playing roles regulating the observed sex differences in both cell abundance and gene expression (*Gegenhuber and Tollkuhn, 2020*). Together, our data highlights two cell types that are enriched for sex-biased gene expression in both a promiscuous and monogamous species, suggesting that these cell types may function in highly conserved, sex-specific pathways.

Finally, we performed a gene enrichment analysis on all sex-biased genes, compared to a background of expression-matched non-sex-biased genes (Methods). Despite the rapid turnover of sex-biased genes across species, we found sex-biased genes to be overwhelmingly enriched in

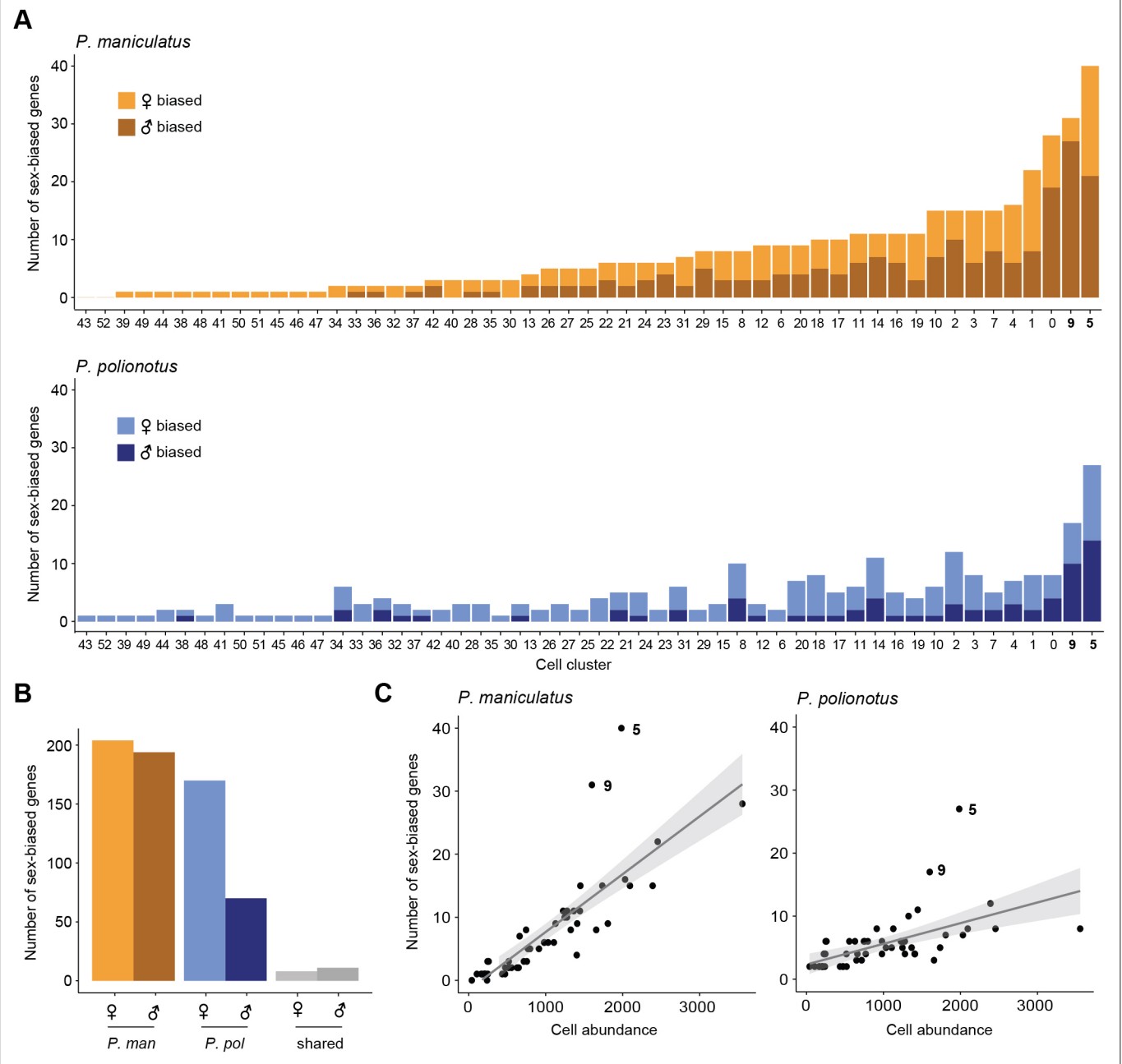

**Figure 5.** Differential gene expression across sex. (**A**) Barplots of the number of sex-biased genes for each cell cluster (bars) in *P. maniculatus* (top) and *P. polionotus* (bottom), colored by female- or male-bias. (**B**) Barplot of the number of female- and male-biased genes aggregated across all cell clusters in *P. maniculatus* (orange), *P. polionotus* (blue), or shared across both species (gray). (**C**) Scatter plots of cell cluster abundance (x-axis) and number of differentially expressed (DE) genes (y-axis) identified in each cell cluster (dots) for *P. maniculatus* (left) and *P. polionotus* (right). Line of best fit (gray line) and 95% confidence interval (gray shading) are shown in each scatter plot.

The online version of this article includes the following source data and figure supplement(s) for figure 5:

**Figure supplement 1.** Cell clusters 5 and 9 are enriched for sex-biased genes and sex steroid receptors.

**Figure supplement 2.** Expression levels of sex steroid receptors by species and sex.

**Figure supplement 2—source data 1.** Related to *Figure 5—figure supplement 2*.

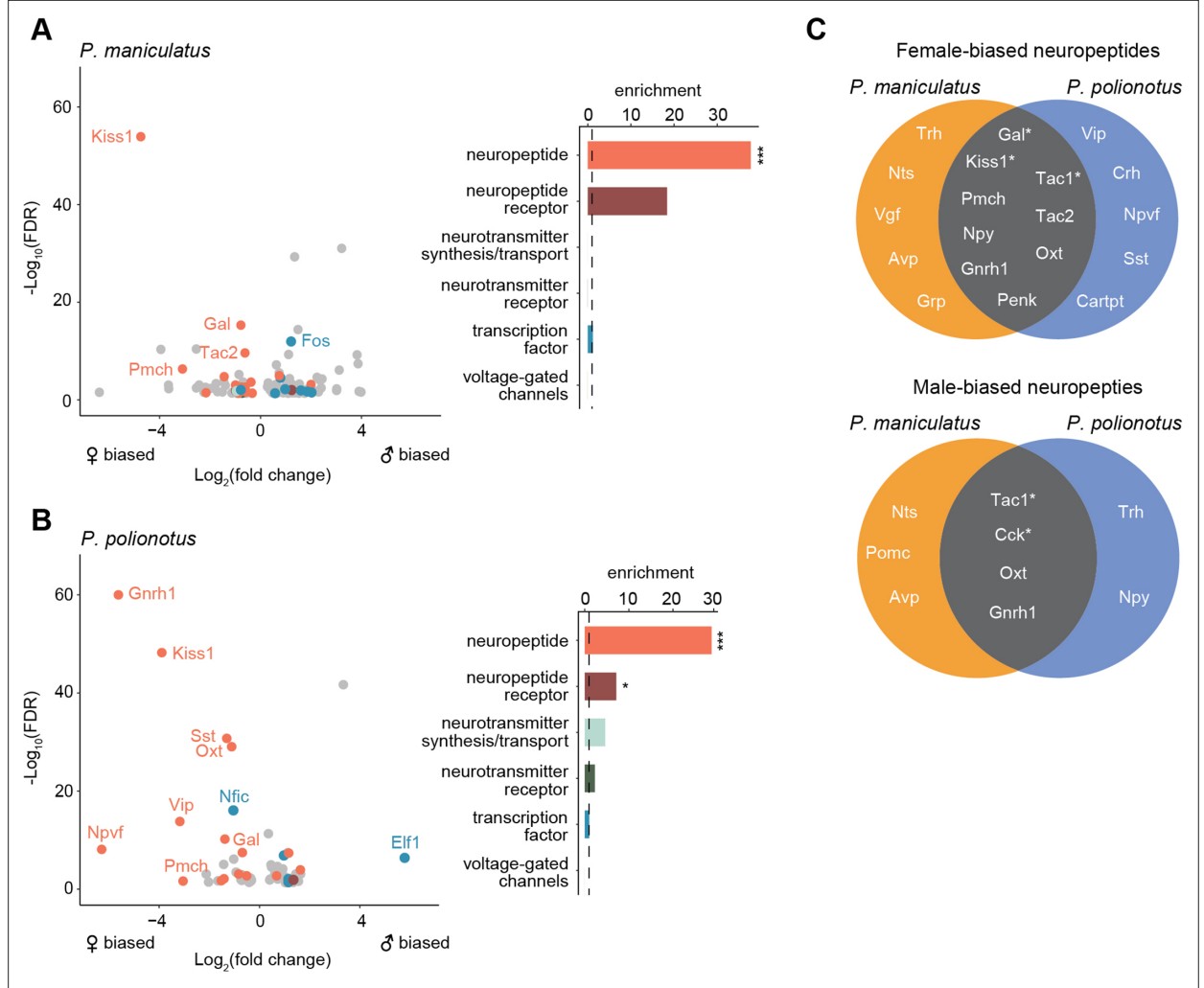

**Figure 6.** Enrichment of neuropeptides in sex-biased genes. (**A**) Left: Volcano plot of log$_2$(fold change) (x-axis) and log$_{10}$(FDR) (y-axis) of sex-biased genes in *P. maniculatus*. Negative values of log$_2$(fold change) indicate female bias and positive values indicate male bias. For genes sex-biased in more than one cell type, the cell type with the highest expression level is shown. Genes are colored by gene categories listed on the right. Select outlier genes are labeled. Right: Barplot of enrichment scores for each gene category. Dotted line at 1.0 indicates no enrichment or depletion. ***FDR <0.001; *FDR <0.05. (**B**) Same as A but for *P. polionotus*. (**C**) Venn diagrams of female- (top) and male-biased (bottom) neuropeptides, categorized by whether they are *P. maniculatus* specific, *P. polionotus* specific, or shared. Some neuropeptides (e.g. *Nts, Tac1, Oxt, Gnrh1*) are female-based in some cell types and male-biased in others. Neuropeptides that share sex-bias *in the same cell type* across species are starred and listed in more detail in.

neuropeptides in both species (29.5 x enrichment, FDR <2e-4 in *P. maniculatus*; 36.8 x enrichment, FDR <2e-4 in *P. polionotus*) (*Figure 6A and B*). Out of 69 annotated neuropeptides in our dataset, 21 were found to be sex-biased in at least one cell cluster in either species (*Figure 6C*). We found neuropeptides were more likely to be female-biased in both species (19 female-biased *vs.* 9 male-biased), although this did not reach statistical significance (Fisher's exact test p=0.06) (*Figure 6C*). We found examples of both species-specific and conserved sex-bias (*Figure 6C*) including 6 instances of neuropeptides sharing the same sex-bias in the same cell type across species (*Table 1*). Therefore, our data suggest that neuropeptides may be playing key roles in sex differences, but that their expression levels appear to be evolutionarily labile, potentially underlying species-specific variation in sex-specific behavior.

Our differential expression analysis identifies striking sex differences in gene expression in the POA. Overall, we found a rapid turnover of sex-biased genes across closely-related species, including a significant reduction in the number of male-biased genes in the monogamous species. Conserved across species, however, we found an enrichment of sex-biased genes expressed in two cell types: cluster 9 (i20:Gal/Moxd1) and cluster 5 (i18:Gal/Tac2). These same cell types were also found to be

**Table 1.** Neuropeptides conserved in sex-bias in same cell types across *P. maniculatus* and *P. polionotus*.

| Neuropeptide | Sex-bias | Cell Type | *P. maniculatus* fold change | *P. maniculatus* FDR | *P. polionotus* fold change | *P. polionotus* FDR |
|---|---|---|---|---|---|---|
| Kiss1 | female | i16:Gal/Th (cluster 20) | 26.0 | 1.2E-54 | 14.9 | 6E-49 |
| Tac1 | female | i18:Gal/Tac2 (cluster 5) | 2.1 | 3.3E-21 | 1.4 | 1.9E-03 |
| Gal | female | i18:Gal/Tac2 (cluster 5) | 1.6 | 3.4E-10 | 1.6 | 3.4E-08 |
| Tac1 | female | cluster 3 | 1.8 | 2.0E-09 | 1.5 | 8.9E-04 |
| Tac1 | male | i20:Gal/Moxd1 (cluster 9) | 3.5 | 1E-30 | 1.5 | 9.8E-03 |
| Cck | male | i20:Gal/Moxd1 (cluster 9) | 1.7 | 1.08E-05 | 2.3 | 7.5E-16 |

differentially abundant across sexes in both species, further underscoring their potential significance in sex-specific biology. Finally, our data suggest that neuropeptides may be a central class of genes mediating sex differences in the brain.

## Discussion

How the evolution of neuronal traits, such as cell type abundance and neuronal gene expression, contributes to variation in innate social behaviors is poorly understood. Here, we explored this question by performing single-nucleus RNA-sequencing in the POA of males and females of a monogamous, biparental species of *Peromyscus* and a species that is promiscuous and exhibits maternal-only care. We focused on the POA as it is a region known to be important for mating and parenting behaviors, and took advantage of an existing *Mus* POA single-cell atlas to assign homology between *Peromyscus* cell clusters and functionally characterized cell types.

### Differential abundance of parenting-related cell types

Of the four cell types differentially abundant across species, three have obvious ties to parenting behavior: AVP (cluster 34), OXT (cluster 40), and i16:Gal/Th (cluster 20) neurons. The differential abundance we observed in AVP neurons both supports our prior finding that variation in *Avp* levels appears to contribute to a parental nesting behavior difference across our focal species (*Bendesky et al., 2017*), and adds to our understanding of the evolutionary mechanisms involved. *Bendesky et al., 2017* previously used Quantitative Trait Locus (QTL) analysis to link parental nest building to two genomic loci, one of which encompasses the *Avp* locus. With bulk RNA-sequencing, *Bendesky et al., 2017* found *Avp* was 3 x more highly expressed in the promiscuous, less parental species (*P. maniculatus*), and further experimentally showed that an increase in intracerebroventricular Avp in *Peromyscus* inhibits parental nesting behavior. The genetic linkage between the *Avp* locus and parental nesting was partially attributed to a *cis*-effect that leads to differential expression across species, though this *cis*-effect changes expression by ~1.1 x and does not explain the full difference in expression across species.

In our study, we found AVP *neuron count* was ~2 x higher in the promiscuous *P. maniculatus*, suggesting that a difference in neuron number may additionally contribute to *Avp* expression and behavioral differences. Additionally, we observed 1.8 x higher expression of *Avp* within *Avp* neurons of *P. maniculatus* compared to *P. polionotus*, corroborating the *cis*-effect found by *Bendesky et al., 2017*. Together, the combination of expression difference and abundance difference better matches the full 3 x difference observed in bulk RNA-sequencing. Our data proposes that evolution may be modulating behavior by changing expression levels through both local, *cis*-regulatory pathways and upstream pathways in neurogenesis and neurodevelopment.

We also found the OXT neuron number to be ~3 x higher in the promiscuous species. The direction of differential abundance is opposite to what may be expected given oxytocin's known roles in pair bonding and monogamy (*Froemke and Young, 2021*). The species difference in oxytocin neurons

may simply be due to the shared developmental lineage between AVP and OXT neurons, which are formed from the same precursor cells and only differentiate from each other later in embryogenesis (*Nakai et al., 1995*; *Schonemann et al., 1995*). Our data suggests that genetic changes upstream of AVP/OXT differentiation have evolved across our focal species. Still, more work needs to be done to understand if the OXT abundance difference leads to behavioral changes, or if, perhaps, they have been compensated for through other changes such as in expression and/or spatial distribution of the oxytocin receptor.

Finally, we find a subtype of galanin neurons, i16:Gal/Th (cluster 20), to be more abundant in monogamous species. Galanin neurons of the MPOA were first reported to be specifically activated by parental care behavior in *Mus* (*Wu et al., 2014*), a finding that has since been replicated in cichlids (*Butler et al., 2020*) and frogs (*Fischer et al., 2019*). These findings suggest galanin has an evolutionarily conserved role in parenting behavior. In a study of biparental poison frogs (*Ranitomeya imitator*), frogs that performed parental behavior in a laboratory assay had significantly more galanin neurons than non-parental frogs, independent of sex (*Fischer et al., 2019*). This study and ours find a correlation between galanin neuronal number and increased propensity for parental care behavior, both within and across species. However, further investigation is required to understand whether these observations are causally linked.

## Two cell types play important roles in sex-specific biology

Across sexes, we found two cell types, i20:Gal/Moxd1 (cluster 9) and i18:Gal/Tac2 (cluster 5), that were sex-biased in cell abundance and had an enrichment of sex-biased genes across both *Peromyscus* species. i20:Gal/Moxd1 (cluster 9) is known to be more abundant in males across many mammalian species in two hypothalamic regions: BNST and SDN-POA (see *Campi et al., 2013* for an overview). However, our immunohistochemistry experiments found the male-bias in abundance to be significantly reduced in the SDN-POA of the monogamous species, though conserved across species in the BNST.

Our findings mirror a previous study comparing sister species of polygamous and monogamous voles, which found the volume of the SDN-POA to be sexually dimorphic in the polygamous species, but not in the monogamous species (*Shapiro et al., 1991*). Another study in *P. californicus*, an independent lineage of monogamous *Peromyscus*, found male-bias in the volume of SDN-POA, but noted that the volume difference was half of what had been reported in other, promiscuous rodents (*Campi et al., 2013*). In contrast, the male-bias found in the volume of the BNST was consistent with fold-differences reported in other rodents (*Campi et al., 2013*). Together, these findings suggest that (1) sex differences in the neuronal number of the BNST and SDN-POA can be independently evolving, even within transcriptionally similar cell types, and (2) sex differences in the neuronal number of the SDN-POA, but not BNST, may be co-evolving with monogamy.

Much less is known about the second sexually dimorphic cell type, i18:Gal/Tac2 (cluster 5). In *Mus*, it is activated by mating behaviors (*Moffitt et al., 2018*) and also reported to be enriched in estrogen receptor (*Esr1*) expression and sex-biased genes (*Kaplan et al., 2024*). Thus, this cell type appears to play conserved sex-specific roles, possibly in mating and copulatory behaviors.

## Sexual dimorphism in gene expression is reduced in monogamous species

Finally, we found significantly fewer sex-biased genes in the monogamous species compared to the promiscuous species. A prevailing hypothesis, first proposed by *Darwin, 1871*, suggests that males of promiscuous species have a larger variance in reproductive success compared to those of monogamous species, leading to greater intrasexual competition and more extreme sexual dimorphisms. Correlations between monogamy and reduced sexual dimorphism have been repeatedly observed in morphological traits such as body size and animal ornamentation (e.g. coloration or plumage) across diverse taxa including primates, birds, fish, and insects (*Andersson, 1994*; *Kvarnemo, 2018*; *Kraaijeveld et al., 2007*). However, much less is known about whether the reduction of sexual dimorphism extends to molecular phenotypes such as gene expression. Work by us and others previously failed to find consistent genome-wide patterns in bulk RNA-sequencing data (*Mishra et al., 2024*; *Kautt et al., 2024*). The striking reduction we see in our snRNA-seq data highlights the importance of

quantifying gene expression at the level of specific cell types to better understand evolution of gene expression and its relationship with organismal-level phenotypes.

Across our study species, there is also a reduction of sexual dimorphism in behavior, with the emergence of biparental care in the monogamous species (in contrast to maternal-only care exhibited by the promiscuous species). Whether the reduced sexual dimorphism in neuronal gene expression may be actually mediating the reduced sexual dimorphism in social behavior is an interesting avenue for future research.

## Conclusion

This study begins to characterize molecular differences in a region of the brain responsible for mating and parenting behaviors across two species with divergent mating systems. Overall, we find cell abundance and gene expression changes across sex and species, both of which contribute to differential levels of neuronally-important proteins. Though not explored here, changes in cell abundance may additionally accompany neuronal circuitry changes that affect behavior. Finally, we find intriguing differences in neuropeptidergic cell types as well as cell types expressing high levels of sex steroid hormone receptors, implicating two key classes of signaling molecules – neuropeptides and sex hormones – as important evolutionary dials for modulating species and sex-specific behavior.

We note that while our two focal species most obviously differ in mating and parenting behaviors, they also differ in other behaviors including thermoregulatory behavior and burrowing behavior (*Bedford and Hoekstra, 2015*). Additionally, the POA is known to regulate behaviors other than mating and parenting behavior, including sleep and thermoregulation. Therefore, further experimental characterization is required to more precisely understand the behavioral consequences of the species differences we observed in the POA, and to reveal the precise nature of how genes can encode for the substrates needed for generating variation in innate behavior.

## Methods

### Key resources table

| Reagent type (species) or resource | Designation | Source or reference | Identifiers | Additional information |
|---|---|---|---|---|
| Biological sample (*Peromyscus maniculatus bairdii*) | Dissections of preoptic area | Peromyscus Genetic Stock Center (Columbia, SC, USA) | | Freshly isolated from *Peromyscus maniculatus bairdii* |
| Biological sample (*Peromyscus polionotus subgriseus*) | Dissections of preoptic area | Peromyscus Genetic Stock Center (Columbia, SC, USA) | | Freshly isolated from *Peromyscus polionotus subgriseus* |
| Antibody | anti-Avp (Rabbit polyclonal) | ImmunoStar | Cat# 20069 RRID:AB_572219 | IHC (1:4000) |
| Antibody | anti-Oxt (Mouse monoclonal) | EMD Millipore | Cat# MAB5296 | IHC (1:2000) |
| Antibody | anti-Calb1 (Mouse monoclonal) | Millipore Sigma | Cat# C9848 | IHC (1:1000) |
| Commercial assay or kit | 10 X Genomics 3' V3 Chip | 10 x Genomics | | Nuclei were loaded at a target of 20,000 nuclei/run |
| Software | CellRanger v5.0.0 | 10 x Genomics | RRID:SCR_023221 | |

### Animal husbandry

*P. maniculatus bairdii* and *P. polionotus subgriseus* animals were originally acquired from the Peromyscus Genetic Stock Center (Columbia, SC, USA). We weaned animals at 23 d of age into single-sex, same-species groups. We maintained animals on a 16 hr:8 hr light:dark cycle at 22 °C, housed them in standard mouse cages with corncob bedding, and provided them with food and water ad libitum. Animal husbandry and experimental procedures were approved by the Harvard University Faculty of Arts and Sciences Institutional Animal Care and Use Committee (protocol 27-15-3).

### POA dissection

Virgin animals between ages 60 and 103 d were euthanized by $CO_2$ and brains were rapidly dissected, flash-frozen in Tissue-Tek Optimal Cutting Temperature Compound, and stored at –70 °C. The POA was dissected on a cryostat (Leica CM3050 S) at –20 °C. Brains were sectioned coronally until reaching the widening of the anterior commissure. The dorsal and lateral regions of the brain were removed so

that only a ~2.5 mm × 2.5 mm region that contained the POA remained (*Figure 1B*). Five 150 µm thick sections were then made to capture the POA. Sections were placed in a 2 mL tube which was kept at –70 °C until the day of nuclei extraction.

## Nuclei extraction and 10x library construction

Nuclei were isolated as described in *Kamath et al., 2022*. Briefly, dissected frozen tissue was placed in a single well of a 6-well plate, with 2 ml of extraction buffer. Mechanical dissociation was performed by trituration using a P1000 pipette, pipetting 1 ml of solution slowly up and down, 20 times. Trituration was repeated five times, with 2 min of rest between each time. The entire volume of the well was then passed twice through a 26 gauge needle into the same well. The tissue solution was transferred into a 50 ml Falcon tube filled with 30 ml wash buffer and spun in a swinging-bucket centrifuge for 10 min at 600 g and 4 °C. Following spinning, the majority of supernatant was discarded, leaving 500 µl of tissue solution. DAPI was added (1:1,000) and debris was removed from nuclei by FACS before nuclei were loaded into a 10 X Genomics 3' V3 Chip. A detailed protocol can be found at dx.doi.org/10.17504/protocols.io.bi62khge.

## Single-nucleus RNA-sequencing

Our snRNA-seq data consists of two experiments: a pilot experiment that aimed to obtain a single replicate of 10,000 nuclei which was sequenced on a NovaSeq S1 lane (rep01), and a full experiment that aimed to obtain five replicates of 20,000 nuclei which were sequenced on a NovaSeq S4 flowcell (rep02 - rep06). Each replicate contained pooled nuclei from a male and female from each of the *P. maniculatus* and *P. polionotus* species (*Figure 1C*). Our final dataset captured 105,647 nuclei across six replicates sequenced to a depth of 1623–2320 median UMI counts/cell (*Supplementary file 1*). The 10 x library construction and sequencing were performed by the Harvard Bauer Core Facility.

## Bulk RNA-sequencing and genotyping

For our pilot study (rep01), we dissected out a portion of the hypothalamus posterior to the POA from each animal for bulk RNA-sequencing. Samples were collected in Trizol (Invitrogen), and total RNA was extracted using the RNeasy mini kit (QIAGEN). Poly(A) enrichment, directional mRNA library preparation, and paired-end sequencing of 150 base pair reads on a NovaSeq 6000 were performed by Novogene. Samples were sequenced to a depth of ~35 million read pairs. Reads were mapped to their respective genomes, Pman_2.1.3 (GCA_003704035.1) or Ppol_1.3.3 (GCA_003704135.2), using STAR with default parameters (*Dobin et al., 2013*). Variants were then called using the joint genotyping procedure recommended by the Genome Analysis Toolkit (GATK) (*Van der Auwera et al., 2013*).

## Whole genome sequencing (WGS) and genotyping

For our full experiment (rep02 - rep06), we performed WGS on animals for genotyping. We extracted genomic DNA from tail tissue using proteinase K digestion followed by the Maxwell RSC (Promega) DNA extraction. We prepared DNA libraries using the Nextera DNA Flex kit with Illumina adapters. All samples were pooled into a single library and sequenced on two NovaSeq S4 lanes with paired-end sequencing of 100 base pair reads. Reads were mapped to their respective genomes using bwa-mem with default parameters (*Li, 2013*). The median read depth for these samples was 267,987,590 mapped reads per sample, which corresponds to ~20 X sequencing coverage across the genome. Variants were called using the GATK procedure described above.

## Sample demultiplexing

To demultiplex pooled scRNA-seq runs, we first used 10 x Genomics CellRanger v5.0.0 with default parameters to map reads to a combined *P. maniculatus* and *P. polionotus* transcriptome. Our in-house transcriptome annotations were generated using Comparative Annotational Toolkit (*Fiddes et al., 2018*) and GENCODE v15 annotations of *Mus musculus* (GRCm38 mm10) as reference. We used the standard CellRanger pipeline to filter out droplet barcodes associated with background noise (*Zheng et al., 2017*). For each droplet that passed the background filter, we preliminarily assigned each droplet to the species for which the majority of reads mapped to (which we refer to as the 'primary' species). Droplets clearly fell into two clusters: one cluster that represents singlets in which

a small percentage of reads incorrectly maps to the secondary species and a second, sparser cluster of droplets that represent multiplets in which genetic material from two species were truly captured (*Figure 1—figure supplement 1*). To delineate between the two clusters, we examined the number of reads mapping to the primary genome and binned droplets in increments of 500 reads. Within each bin, we then plotted the distribution of reads mapping to the *secondary* genome. We fit this distribution with a mixture model of two Gaussians using normalmixEM (*Benaglia et al., 2010*) and identified the point at which the two distributions cross (*Figure 1—figure supplement 1*). We then found the line of best fit through all intersection points across the bins and used this line as the threshold between singlets and multiplets (*Figure 1—figure supplement 1*). Within each species, we next used demuxlet (*Kang et al., 2018*) to infer sample identity based on the known sample genotype data (*Figure 1—figure supplement 2*). Demuxlet additionally identifies multiplets when mixtures of sample genotypes are detected.

For each 10 x run, we expected our detectable multiplet rate to be ~9% (*Genomics, 2022*) and our assigned detectable rate ranged from 8–13% across our replicates (*Figure 1—figure supplement 2*). We found that our assigned singlets were evenly distributed across the four samples (*Figure 1—figure supplement 3A*) and the distribution of the number of genes detected per droplet was similar across samples and replicates (*Figure 1—figure supplement 3B*). Finally, when we checked the expression level of the female-specific gene, *Xist*, we found that female-assigned cells had significantly higher *Xist* read counts than male-assigned cells and the median read counts of *Xist* across male-assigned cells was 0 (*Figure 1—figure supplement 3C*). After sample demultiplexing, we used CellRanger v5.0.0 to map reads from each nucleus to the transcriptome of its assigned species and used these read counts for all subsequent analyses.

## Cell type clustering

All normalization, data integration, and clustering of scRNA-seq data were performed with Seurat v4.1.3 (*Hao et al., 2021*). Within species, we normalized our scRNA-seq data using sctransform (*Hafemeister and Satija, 2019*; *Choudhary and Satija, 2022*) and then integrated data across species using Seurat's integration workflow with default parameters. Briefly, Seurat learns a conserved gene correlation structure between two data sets using canonical correlation analysis and identifies pairwise correspondences between single nuclei across datasets in order to transform data sets into a shared space (*Stuart et al., 2019*). We created UMAPs using the runUMAP function, and clustered the integrated data by using the FindNeighbors function.

## Homology mapping of inhibitory and excitatory cell types

We defined each cell cluster and all nuclei within that cluster as inhibitory or excitatory based on higher expression levels of *Gad1* and *Gad2* (inhibitory) or *Slc17a6* (excitatory) (*Figure 1—figure supplement 7*). We then mapped inhibitory and excitatory cell type labels from the *Mus* POA atlas to inhibitory and excitatory nuclei, respectively, using Seurat's label transfer workflow with default parameters (*Stuart et al., 2019*). Briefly, Seurat first identifies pairwise correspondences between single nuclei across datasets ('anchors') and then 'transfers' cell type labels by weighting the cell type identities of each anchor with their distance to the nuclei at hand. For each nuclei, this procedure calculates the probability that it belongs to each cell type label and assigns the cell type label which received the highest probability.

To find cluster-to-cell-type mappings, we calculated the proportion of each cluster that was assigned to each *Mus* cell type label (*Figure 1—figure supplement 8*). We removed *Mus* cell type labels that lacked any notable marker genes and that indiscriminately mapped onto many of our clusters (e1:Glut, i1:Gaba, i7:Gaba, i9:Gaba, i11:Gaba, i13:Gaba, i12:Gaba, i15:Gaba, i39:Gaba) (*Figure 1—figure supplement 8*). We then retained only cell clusters and *Mus* cell type labels if 15% of a cluster or more were predicted to belong to the same *Mus* cell type label. The cutoff value was chosen empirically after examining clusters where homology to the *Mus* atlas that could be established by marker genes. The proportion of these clusters that mapped to *Mus* labels ranged from 17.3% (cluster 15 - i14:Avp/Cck) to 92.6% (cluster 38 – i4:Gaba/Mylk), and therefore a 15% cutoff was chosen. This resulted in 17 excitatory clusters mapping onto 11 *Mus* excitatory cell type labels and 17 inhibitory clusters mapping onto 18 *Mus* inhibitory cell type labels (*Figure 1E and F*).

## Differential abundance of cell clusters

Differential abundance analysis was conducted with edgeR (*Robinson et al., 2010*). After clustering our neuronal snRNA-seq data, we created a count matrix of the number of nuclei within each cluster that belonged to each sample. Cluster sizes were normalized using TMM normalization, which relies on the assumption that there are minimal differences in cluster sizes across samples (*Robinson and Oshlack, 2010*). We used the estimateDisp function with default parameters to estimate the dispersions across cell abundance counts and calcNormFactors function with default parameters to control for the differential numbers of total neurons profiled per sample. We then fit a generalized linear model with glmQLFit function and a design that included replicate, species, and sex as covariates. Finally, we used glmQLFTest to test for significant coefficients and corrected p-values with an FDR correction (*Benjamini and Hochberg, 1995*).

## Immunohistochemistry and cell counting

We transcardially perfused mice with ice-cold 1 x phosphate-buffered saline (PBS) with heparin (100 U/mL) and then with 4% paraformaldehyde in PBS. Brains were dissected out, postfixed for 24 hr at 4 °C, cryopreserved in 30% sucrose, and stored at −70 °C until subsequent use. To stain for protein, we sectioned brains at 30 µm. We aimed to capture the entire hypothalamus and collected sections beginning at the appearance of the corpus callosum until the appearance of the thalamus, keeping every second section (48 sections spanning ~3 mm). We blocked free-floating sections for 1 hr in 10% normal goat serum (NGS) and 0.5% Triton-X in PBS. We incubated sections overnight with either rabbit anti-Avp (1:4000, ImmunoStar 20069), mouse anti-Oxt (1:2000, EMD Millipore MAB5296), or mouse anti-Calb1 antibody (1:1000, Millipore Sigma C9848) and 10% NGS in PBS. We used either donkey anti-rabbit Alexa 647 antibody (1:1000, Thermo Fisher Scientific A31573) or donkey anti-mouse Alexa 546 (1:1000, Thermo Fisher Scientific A11003) for secondary detection and mounted tissues with DAPI Fluoromount-G (SouthernBiotech, 0100–20). Slides were imaged on an AxioScan. Z1 slide scanner (Zeiss).

Following imaging, we exported images to .tif format and renamed files so that cell counters were blinded to the species and sex. We counted cells using a custom Fiji/ImageJ macro. Briefly, we manually outlined a region of interest (ROI) (e.g. PVN, SON, etc.). Within the ROI, we binarized the image using the 'Moments' method (*Tsai, 1985*) for calculating the optimal threshold. We then used the watershed segmentation algorithm to separate touching particles. Finally, we used the Analyze Particles function to count the number of particles inside the ROI.

*P. maniculatus* brains are ~15% larger than *P. polionotus* brains by weight which could impact species comparisons of neuron numbers. However, because we were confirming species differences in AVP and OXT neurons, which were more abundant in *P. polionotus*, we did not normalize by brain size for those analyses.

## Differential expression analysis

To obtain pseudobulked data, gene counts were summed across all cells belonging to the same cell cluster and sample resulting in a matrix of 30,704 genes by 24 samples by 53 cell clusters. Differential expression analysis was then performed with edgeR (*Robinson et al., 2010*), using the same procedure described above for differential abundance analysis. For differential expression analysis across species, we fit a generalized linear model with a design that included replicate, sex, and species as covariates. For differential expression analysis across sex, we performed edgeR analysis separately for each species and fit a generalized linear model that included only replicate and sex as covariates. We accounted for multiple hypothesis testing with an FDR correction (*Benjamini and Hochberg, 1995*) and used an FDR cutoff of 0.05 to define the significant differential expression.

## Gene enrichment analysis

Gene lists were manually curated by using the Gene Ontology Resource (*Blake et al., 2013*) and are available in *Supplementary file 5*. To test for enrichments of each gene category, we compared foreground genes (i.e. a significantly differentially expressed gene) to background genes that were matched in expression level to control for the fact that genes that are more highly expressed are more accurately quantified and thus more likely to be identified as differentially expressed. We created backgrounds with the following procedure: We binned all genes by their expression level in bins of 0.5

$\log_{10}$(counts per million). For each gene in our foreground, we randomly chose a gene from our entire dataset from the same expression bin. If the same gene was DE in multiple cell types, we matched the highest expression level. To obtain an empirical *p*-value, we repeated this procedure 5000 times and calculated the number of times a background list contained equal or higher numbers of foreground genes within a gene category, and we accounted for multiple hypothesis testing of several gene categories using an FDR correction. We calculated the enrichment score by dividing the number of foreground genes by the mean number of background genes (across 5000 random samples) within each gene category.

## Acknowledgements

We thank Dhananjay Bambah-Mukuu and Harris Kaplan for advice on POA dissection; Charles Vanderburg and Naeem Nadaf for advice on single-nucleus extraction; and Doug Richardson for advice on cell counting. JC was supported by the Harvard Data Science Initiative and the National Institutes of Health (K99 GM146243-01). SRE and HEH were supported by the Howard Hughes Medical Institute.

## Additional information

### Funding

| Funder | Grant reference number | Author |
| --- | --- | --- |
| National Institutes of Health | GM146243-01 | Jenny Chen |
| Harvard Data Science Initiative, Harvard University | | Jenny Chen |
| Howard Hughes Medical Institute | | Jenny Chen<br>Phoebe R Richardson<br>Christopher Kirby<br>Sean R Eddy<br>Hopi E Hoekstra |

The funders had no role in study design, data collection and interpretation, or the decision to submit the work for publication.

### Author contributions

Jenny Chen, Conceptualization, Resources, Data curation, Software, Formal analysis, Supervision, Funding acquisition, Validation, Investigation, Visualization, Methodology, Writing – original draft, Project administration, Writing – review and editing; Phoebe R Richardson, Christopher Kirby, Data curation; Sean R Eddy, Hopi E Hoekstra, Supervision, Funding acquisition, Writing – review and editing

### Author ORCIDs

Jenny Chen ⬤ https://orcid.org/0000-0002-6664-2597
Phoebe R Richardson ⬤ https://orcid.org/0009-0008-5650-1556
Christopher Kirby ⬤ https://orcid.org/0000-0001-7292-9044
Sean R Eddy ⬤ https://orcid.org/0000-0001-6676-4706
Hopi E Hoekstra ⬤ https://orcid.org/0000-0003-1431-1769

### Ethics

All experimental procedures were approved by the Harvard University Institutional Animal Care and Use Committee (IACUC protocol no. protocol 27-15). The animal housing facility in which these tests were performed maintains full AAALAC accreditation.

Reviewer #1 (Public review): https://doi.org/10.7554/eLife.103109.3.sa1
Reviewer #2 (Public review): https://doi.org/10.7554/eLife.103109.3.sa2
Reviewer #3 (Public review): https://doi.org/10.7554/eLife.103109.3.sa3
Author response https://doi.org/10.7554/eLife.103109.3.sa4

## Additional files

### Supplementary files

Supplementary file 1. 10 x sequencing statistics.

Supplementary file 2. Cell abundance of all clusters across all replicates.

Supplementary file 3. Differential expression across species.

Supplementary file 4. Differential expression across sex.

Supplementary file 5. Gene lists.

MDAR checklist

### Data availability

All sequencing data has been deposited in GEO under accession number GSE272719. Analysis scripts are available at GitHub (copy archived at *Chen, 2024*).

The following dataset was generated:

| Author(s) | Year | Dataset title | Dataset URL | Database and Identifier |
|---|---|---|---|---|
| Chen J, Richardson PR, Kirby C, Eddy SR, Hoekstra HE | 2024 | Cellular evolution of the hypothalamic preoptic area of behaviorally divergent deer mice | https://www.ncbi.nlm.nih.gov/geo/query/acc.cgi?acc=GSE272719 | NCBI Gene Expression Omnibus, GSE272719 |

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
