## [Editor Report · eLife Assessment]

This **important** study identifies species- and sex-specific neuronal cell types and gene expression in the preoptic area (POA) to help understand the evolutionary divergence of social behaviors. The evidence from single-nucleus RNA sequencing and immunostaining is **compelling** and suggests that cellular differences in the POA may contribute to behavioral variations such as mating and parental care that are apparent in two closely related deer mouse species. These rich observations provide an entry point for future hypothesis-driven experiments to demonstrate a causal role for these populations in sex- or species-variable behaviors in vertebrates. These data will be a resource that is of value to behavioral neuroscientists.

---

## [Referee Report · Reviewer #1 (Public review)]

(1) Summary of the Paper:

This paper by Chen et al. examines the cellular composition and gene expression of the hypothalamic medial preoptic area (MPOA) in two closely related deer mouse species (P. maniculatus and P. polionotus) that exhibit distinct social behaviors. Through single-nucleus RNA sequencing (snRNA-seq), Chen et al., identify sex- and species-specific neuronal cell types that likely contribute to differences in mating and parental care. By comparing monogamous and promiscuous species, the study provides insights into how neuronal diversity and gene expression changes in the MPOA might underlie the evolution of social behaviors.

(2) Strengths of the Paper:

The paper excels in several areas. First, the data presentation is clear and well-organized, making the complex findings easy to follow. The writing is straightforward and highly accessible, which enhances the overall readability. The experimental design is innovative, particularly in how they combined samples from different species into the same dataset and then used post-hoc identification to distinguish cell types by species. This dramatically controls for potential batch effects in my opinion. Additionally, the authors contextualize their findings within the framework of previously published studies on *Mus musculus*, providing a strong comparative analysis that enhances the significance of their work.

(3) Weaknesses of the Paper:

The major limitation of the study is the absence of causal experiments linking the observed changes in MPOA cell types to species-specific social behaviors. While the study provides valuable correlational data, it lacks functional experiments that would demonstrate a direct relationship between the neuronal differences and behavior. For instance, manipulating these cell types or gene expressions in vivo and observing their effects on behavior would have strengthened the conclusions, although I certainly appreciate the difficulty in this, especially in non-musculus mice. Without such experiments, the study remains speculative about how these neuronal differences contribute to the evolution of social behaviors.

---

## [Referee Report · Reviewer #2 (Public review)]

Summary:

The authors report several interesting species and sex differences in cell type expression that may relate to species differences in behavior. The differential cell type abundance findings build on previously observed species/sex differences in behavior and brain anatomy. These data will be a valuable resource for behavioral neuroscientists. These findings are important but the manuscript goes too far in attributing causal influences to differences in behavior. A second important problem is that dissections used for the sequencing data include other neuropeptide-rich areas of the hypothalamus like the PVN. Although histology is included, the results into the main manuscript often do not include the mPOA making it hard to know if species/sex differences are consistent across different hypothalamic regions. The manuscript would benefit from more precise language.

Strengths:

The data are novel because cell-type atlases are available for only a few species.

The authors have clearly defined appropriate steps taken to obtain trustworthy estimations of cell type abundance. Furthermore, the criteria for each cell type assignment was described in a way for readers to easily replicate. The rigor in comparing cell abundance provides convincing evidence that these species have differences in MPOA cellular composition.

The authors have a good explanation for why 19 of the 53 neuron clusters were not classified (possible Mus/Peromyscus anatomical differences, some cell types don't have well-defined transcriptional profiles)

Validated findings with histology.

---

## [Referee Report · Reviewer #3 (Public review)]

Summary:

The authors performed snRNA-seq in the pre-optic area (POA), a heterogeneous brain region implicated in multiple innate behaviors, comparing two species of Peromyscus mice that possess strikingly different parenting behaviors. P. polionotus show high levels of parental care from both sexes of parent, and P. maniculatus show lower levels of care, predominantly displayed by dams rather than sires. The overall goal of understanding the genomic basis of behavioral variation is significant and of broad interest and comparative studies in POA in these two species is an excellent approach to tackle this question. The authors correctly point out that existing studies largely compare species that are highly divergent, such as mice and humans, which confounds the association of specific neuronal populations or gene expression patterns with distinct behaviors. They identify neuronal populations with differential abundance between species and sexes, and additionally report sex and species differences in gene expression within each transcriptomic cell type. Their cell type classification is aided by mapping their Peromyscus cells onto a previously existing POA single cell dataset generated in lab mice. The detection and validation of previously observed sex differences in the Gal/Moxd1 cell type, and species differences in Avp expression provides additional support that their data are robust. Importantly, the authors demonstrate reduced sexual dimorphism in the POA of P. polionotus, compared to P. maniculatus, and prior knowledge in rats and mice. This finding suggests a potential neural substrate for the increased parental behavior in P. polionotus.

Strengths:

This is a pioneering comparative snRNA-seq study that provides a roadmap for similar approaches in non-traditional model organisms.

The authors have identified populations that may underlie sex- and species- differences in parenting behavior in rodents.

A significant strength of the manuscript is the histological validation of their most robust marker genes.

Weaknesses:

My primary concern is that the dataset is limited: 52,121 neuronal nuclei across 24 samples, which does not provide many cells per cluster to analyze comparatively across sex and species, particularly given the heterogeneity of the large region dissected, which contains adjacent regions such as the PVN and SCN.

There is no explanation for the finding that there is a female-bias in gene expression across all cell types in P. polionotus.

---

## [Author Response]

The following is the authors’ response to the original reviews.

We thank the reviewers for their thoughtful comments.

Based on their suggestions we will:

(1) Use more accurate language to describe the hypothalamus regions under investigation in this study. While we aimed to primarily investigate the medial preoptic area (MPOA), our dissections and sequencing data in fact capture several regions of the anterior hypothalamus including the anteroventral periventricular (AVPV), paraventricular (PVN), supraoptic (SON), suprachiasmatic nuclei (SCN), and more. We will revise the language in our manuscript to reflect that our study in fact investigates the cellular evolution of the anterior hypothalamus across behaviorally divergent deer mice.

(2) Revise our language to clarify that while our study provides a rich dataset for generating hypotheses about which cell types may contribute to behavioral differences, it does not provide any evidence of causal relationships. We hope to investigate this further in future work.

(3) Clarify specific methodological choices for which reviewers had questions, especially about the hypothalamic regions for which we did histology to validate cell abundance differences and methodological choices related to mapping our cell clusters to *Mus* cell types.

Our responses to each reviewer’s specific comments are below.

**Reviewer #1:**
The major limitation of the study is the absence of causal experiments linking the observed changes in MPOA cell types to species-specific social behaviors. While the study provides valuable correlational data, it lacks functional experiments that would demonstrate a direct relationship between the neuronal differences and behavior. For instance, manipulating these cell types or gene expressions in vivo and observing their effects on behavior would have strengthened the conclusions, although I certainly appreciate the difficulty in this, especially in non-musculus mice. Without such experiments, the study remains speculative about how these neuronal differences contribute to the evolution of social behaviors.

Yes, we agree the study lacks functional experiments. We hope that the dataset is of value for generating hypotheses about how hypothalamic neuronal cell types may govern species-specific social behaviors, and for these hypotheses to be functionally tested by us and others in future work.

**Reviewer #2:**
Some methodology could be further explained, like the decision of a 15% cutoff value for cell type assignment per cluster, or the necessity of a multi-step analysis pipeline for gene enrichment studies.

A 15% cutoff value for cell type assignment was chosen to include all known homology correspondences between our dataset and the *Mus* atlas. For example, i14:Avp/Cck cells from the *Mus* atlas represent Avp cells from the suprachiasmatic nuclei (SCN). Though only 17.3% of cluster 15 maps to i14:Avp/Cck, we know these two clusters correspond based on the expression of *Avp* and additional SCN marker genes in cluster 15 (Supp Fig 6). We will further explain this cutoff in the revised manuscript.

Our gene enrichment study includes a multi-step analysis pipeline because we wanted to control for confounders that may be introduced because of gene expression level. Genes that are more highly expressed are more accurately quantified and thus more likely to be identified as differentially expressed. Therefore, we wanted to test for gene enrichments in our set of DE genes against a background of genes with similar expression levels. We will clarify this motivation in the revised manuscript.

The authors should exercise strong caution in making inferences about these differences being the basis of parental behavior. It is possible, given connections to relevant research, but without direct intervention, direct claims should be avoided. There should be clear distinctions of what to conclude and what to propose as possibilities for future research.

Yes, we agree that we are unable to make direct claims about neuronal differences being the basis of parental behavior. We will revise our language to be clearer about which relationships we are hypothesizing and what we propose as possibilities for future research.

Histology is not performed on all regions included in the sequencing analysis.

We apologize that our language describing the hypothalamic regions included in the sequencing analysis and those included in the histology is unclear. We aimed to dissect the medial preoptic region for the sequencing analysis, but additionally captured parts of the anterior hypothalamus including the paraventricular (PVN), supraoptic (SON), and suprachiasmatic nuclei (SCN), and more. Our histology was performed across the entire hypothalamus and includes all regions included in the sequencing data. We will revise the manuscript to more accurately describe the hypothalamic regions for which we investigated.

**Reviewer #3:**
My primary concern is that the dataset is limited: 52,121 neuronal nuclei across 24 samples, which does not provide many cells per cluster to analyze comparatively across sex and species, particularly given the heterogeneity of the region dissected. The Supplementary table reports lower UMIs/genes per cell than is typically seen as well. Perhaps additional information could be obtained from the data by not restricting the analyses to cells that can be assigned to Mus types. A direct comparison of the two Peromyscus species could be valuable as would a more complete Peromyscus POA atlas.

Our dataset reports ~1,500 genes and ~1,000 UMIs per nuclei which is indeed lower than is typically reported in other single nuclei datasets. Some of this discrepancy is due to a lower quality genome and annotated transcriptome available for *Peromyscus* compared to *Mus musculus*, which results in a lower mapping rate than is typically reported in *Mus* studies. However, our dataset was sufficient to identify known peptidergic cell types (Supp Fig 6) and to map homology to *Mus* cell types for 34 (64%) of our 53 clusters. Additionally, although some of our clusters contain small numbers of cells, our differential abundance analysis accounts for the variance in cell numbers observed across samples and should be robust against any increase in variance due to small numbers. In fact, even differential abundance of very small cell clusters such as oxytocin neurons (cell type 40) was validated by histology.

We would like to clarify that all analyses were performed on all cell clusters, regardless of whether or not they could be assigned homology to a *Mus* cell type. All the cell types that we identified as differentially abundant or contained significant sex differences happened to be cell types for which homology to a *Mus* cell type could be defined. This may arise for a relatively uninteresting reason: cell types that have more distinct transcriptional signatures will be more accurately clustered, leading to more accurate identification of homology as well as more accurate measurements of differential abundance / expression. We will revise language to make this more clear in our manuscript.

In Supplement 7, it appears that most neurons can be assigned as excitatory or inhibitory, but then so many of these cells remain in the unassigned "gray blob" seen in panel 1E. Clustering of excitatory and inhibitory neurons separately, as in prior cited work in Mus POA (refs 31 and 57) may boost statistical power to detect sex and species differences in cell types. Perhaps the cells that cannot be assigned to Mus contain too few reads to be useful, in which case they should be filtered out in the QC. The technical challenges of a comparative single-cell approach are considerable, so it benefits the scientific community to provide transparency about them.

We are not certain about why we are unable to cluster and assign homology to many of our cells (i.e. cells in the unassigned “gray blob”). However, we note that even in the *Mus* atlas, many cells did not belong to obvious clusters by UMAP visualization and that several clusters lacked notable marker genes and were designated simply as “Gaba” and “Glut” clusters. Therefore, it is unsurprising that our own dataset also contains cells that lack the transcriptional signatures needed to be clustered and/or mapped to *Mus* cell types. We do know, however, that the median number of reads/nuclei is uniform across cell clusters and does not explain why some clusters could not be assigned to *Mus*. We will add this information to our revised manuscript.

We do not think that a two-stage clustering (i.e. clustering first by excitatory vs. inhibitory neurons) is expected to gain power to resolve cell types in this case. Excitatory vs. inhibitory neurons are clearly separable on our UMAP (Supp Fig 7) so that information is already being used by our clustering procedure. However, we will explore this further in our revised manuscript to see if doing so will boost statistical power.

The Calb1 dimorphism as observed by immunostaining, appears much more extensive in P. maniculatus compared to P. polionotus (Figures 3 E and F). This finding is not reflected in the counts of the i20:Gal/Moxd1 cluster. The use of Calb1 staining as a proxy for the Gal/Moxd1 cluster would be strengthened if the number of POA Calb1+ neurons that are found in each cluster was apparent. There may be additional Calb+ neurons in the cells that are not annotated to a Mus cluster. This clarification would add support to the overall conclusion that there is reduced sexual dimorphism in P. polionotus.

From the *Mus* MPOA atlas (which includes both single-cell sequencing data and imaging-based spatial information), it is known that the i20:Gal/Moxd1 cluster comprises sexually dimorphic cells that make up both the BNST and the SDN-POA. These sexually dimorphic cells are well-studied and known to be marked by *Calb1*, which we used in immunostaining as a proxy for i20:Gal/Moxd1.

However, we would like to clarify that in our study, the immunostaining of Calb1+ neurons and the sequencing counts of the i20:Gal/Moxd1 cluster are not completely reflective of each other because our sequencing dataset only captured the ventral portion of the BNST. Therefore our i20:Gal/Moxd1 counts contain a combination of some Calb1+ BNST cells and likely all Calb1+ SDN-POA cells and is difficult to interpret on its own. Our histology, however, covers the entire hypothalamus and is more reliable for identifying sex and species differences in each region. We will clarify this in the revised manuscript.

The relationship between the sex steroid receptor expression and the sex bias in gene expression would be improved if the sex bias in sex steroid receptor expression was included in Supplementary Figure 10.

We will include this in the revised manuscript.

There is no explanation for the finding that there is a female bias in gene expression across all cell types in P. polionotus.

We also find this observation interesting but don’t have a good explanation for why at this point. We plan to follow this up in future work.